# Graphene Oxide Nanosheets for Localized Hyperthermia—Physicochemical Characterization, Biocompatibility, and Induction of Tumor Cell Death

**DOI:** 10.3390/cells9030776

**Published:** 2020-03-23

**Authors:** Malgorzata J. Podolska, Alexandre Barras, Christoph Alexiou, Benjamin Frey, Udo Gaipl, Rabah Boukherroub, Sabine Szunerits, Christina Janko, Luis E. Muñoz

**Affiliations:** 1Department of Internal Medicine 3—Rheumatology and Immunology, Universitätsklinikum Erlangen, Friedrich-Alexander University (FAU) Erlangen-Nürnberg, 90154 Erlangen, Germany; gosiapodolskao2@wp.pl; 2Univ. Lille, CNRS, Centrale Lille Univ. Polytechnique Hauts-de-France, UMR 8520-IEMN, F-59000 Lille, France; alexandre.barras@univ-lille.fr (A.B.); rabah.boukherroub@iemn.univ-lille1.fr (R.B.); sabine.szunerits@univ-lille1.fr (S.S.); 3Department of Otorhinolaryngology, Head and Neck Surgery, Section of Experimental Oncology and Nanomedicine (SEON), Else Kröner-Fresenius-Stiftung-Professorship, Universitätsklinikum Erlangen, 91054 Erlangen, Germany; christoph.alexiou@uk-erlangen.de (C.A.); christina.janko@uk-erlangen.de (C.J.); 4Department of Radiation Oncology, Universitätsklinikum Erlangen, Friedrich-Alexander-University (FAU) Erlangen-Nürnberg, 90154 Erlangen, Germany; benjamin.frey@uk-erlangen.de (B.F.); udo.gaipl@uk-erlangen.de (U.G.)

**Keywords:** graphene oxide, hyperthermia, radiotherapy, cell death, infrared, hemocompatibility

## Abstract

Background: The main goals of cancer treatment are not only to eradicate the tumor itself but also to elicit a specific immune response that overcomes the resistance of tumor cells against chemo- and radiotherapies. Hyperthermia was demonstrated to chemo- and radio-sensitize cancerous cells. Many reports have confirmed the immunostimulatory effect of such multi-modal routines. Methods: We evaluated the interaction of graphene oxide (GO) nanosheets; its derivatives reduced GO and PEGylated rGO, with components of peripheral blood and evaluated its thermal conductivity to induce cell death by localized hyperthermia. Results: We confirmed the sterility and biocompatibility of the graphene nanomaterials and demonstrated that hyperthermia applied alone or in the combination with radiotherapy induced much more cell death in tumor cells than irradiation alone. Cell death was confirmed by the release of lactate dehydrogenase from dead and dying tumor cells. Conclusion: Biocompatible GO and its derivatives can be successfully used in graphene-induced hyperthermia to elicit tumor cell death.

## 1. Introduction

Nanomaterials play a pivotal role in nanomedicine where they have reached substantial advances in a relatively short time. Nonetheless, most nanocomposites lack proper characterization due to still missing regulations. This results in limited knowledge about their potentially toxic properties and does not allow revealing opportunities and possible limitations of their use in medical applications. One of the relatively new nanomaterials explored as nanocarriers in several diagnostic and therapeutic approaches are graphene-based nanomaterials such as graphene oxide (GO) or reduced graphene oxide (rGO) [1].

Graphene oxide is a multifunctional carbon nanomaterial and, due to its excellent and versatile physicochemical properties, has a high potential for the development of platform technologies for cancer therapies. Its surface can be covalently and noncovalently functionalized with anticancer drugs and cancer cell-targeting ligands to improve treatment efficacy. These GO-based nanomaterials have shown to improve the delivery of the drug at the tumor site by the enhanced permeability and retention (EPR) effect or by using active targeting ligands [2]. However, the targeted release of chemotherapeutics alone has also shown some degree of failure [3]. The great photothermal conversion efficacy of graphene nanosheets, notably rGO, makes this material a potential instrument to induce targeted localized hyperthermia as a booster of tumor cell death in combination with radiotherapy [4,5].

Chemo- and radiotherapy often fail to induce cell death due to developed chemo- and radioresistance of the tumor. It has been demonstrated that hyperthermia can sensitize cancerous cells to such treatments [6]. This is possible due to the increase in cell membrane fluidity, which leads to an increased uptake of chemotherapeutics [7]. In addition, hyperthermia promotes improved free radical generation, denaturation of proteins involved in DNA repair, and eradication of the cells in S-phase of the cell cycle [8,9,10,11], contributing to tumor cell killing when the multimodal therapies are applied. The substantial influence of hyperthermia on the sensitization of tumor cells to other therapies additionally has the potential to induce immunogenic cell death. 

Considering the photothermal conversion effect of GO nanosheets and its derivatives, we suggest that those nanomaterials can be utilized to elicit localized Graphene-Induced Hyperthermia (GIHT) when they are injected intravenously into organisms carrying tumors. Before this novel application has a chance to be tested in well-designed preclinical studies evaluating antitumor immunity, the sterility and biocompatibility of GO, rGO, and PEGylated rGO (rGO-PEG) nanostructures must be thoroughly investigated.

In this work, we present a toxicologic description of three types of GO nanomaterials developed for blood stream applications and their interaction with all the components of peripheral blood and the first cells these nanostructures would encounter after arriving at a solid tumor, namely macrophages and neutrophils. Furthermore, we confirm the enhancement of cell death by hyperthermia induced with the reduced form of GO, notably rGO-PEG, due to a better dispersion in biological fluids. 

## 2. Materials and Methods

### 2.1. Formation of Reduced Graphene Oxide (rGO)

GO was purchased from Graphenea (Spain) and used as received. GO aqueous suspension (150 mg, 3 mg/mL) was prepared by sonication and hydrazine hydrate (50 µL, 1.03 mmol) was added to this suspension. The mixture was heated in an oil bath at 100 °C for 24 h over which the reduced GO gradually precipitated out the solution. The product was isolated by filtration over a polyvinylidene fluoride (PVDF) membrane with a 0.45-μm pore size, washed copiously with water (5 × 20 mL) and methanol (5 × 20 mL), and dried in an oven.

### 2.2. Formation of Poly(Ethylene Glycol) Modified Reduced Graphene Oxide (rGO-PEG)

Caboxylic acid-enriched reduced graphene oxide (rGO-COOH) was first synthesized through chemical reaction with chloroacetic acid under strong basic conditions. Briefly, sodium hydroxide (1.4 g) (Sigma-Aldrich, Schnelldorf, Germany) and chloroacetic acid (1 g) (Sigma-Aldrich, Germany) were added to 50 mL of GO (20 mg) aqueous solution and the mixture was subjected to sonication (35 kHz) for 2 h at 80 °C. The reaction was quenched with HCl (20%) (Sigma-Aldrich, Germany) and centrifuged at 4500 rpm 3 times to remove unreacted material. 

The obtained rGO-COOH was dispersed into MilliQ water (1 mg/mL), followed by adding N-(3-dimethylaminopropyl)-N′-ethylcarbodiimide (98%, EDC) (4 mM) (Sigma-Aldrich, Germany) and N-Hydroxysuccinimide (98%, NHS) (4 mM) (Sigma-Aldrich, Germany) and by sonicating for 30 min. To this solution was added NH2-PEG-NH2 (2 mg/mL) (Sigma-Aldrich, Germany), and the reaction was maintained under sonication (35 kHz) for 2 h at room temperature. Thereafter, the mixture was quenched by mercaptoethanol and dialyzed against MilliQ water using 12–14 kDa molecular weight cutoff membrane. The solution was then centrifuged at 4500 rpm for 30 min before use.

### 2.3. Sterility of Graphene Derivatives

GO, rGO, and rGO-PEG nanosheets were streaked on Luria broth (LB) (Sigma-Aldrich, Germany) agar plates in a dilution of 1:200 in ddH_2_O. *Escherichia coli* (strain DH5α) was used as a positive control of bacterial growth. Agar plates were incubated for 72 h at 37 °C. Macrophotographs were taken with a Nikon D700 reflex camera (Nikon, Tokio, Japan) and were processed using Adobe Photoshop CS5. 

### 2.4. Detection of Endotoxin Contamination

The EndoZyme^®^ II kit (Hyglos, Bernried, Germany) was employed according to the manufacturer’s instructions. Endotoxin-free equipment was used. Briefly, graphene derivatives were co-incubated with recombinant Factor C and synthetic fluorogenic substrate for 60 min at 37 °C. Five units/mL lipopolysaccharide (LPS) was added to the nanosheets’ suspensions to exclude absorption of fluorescence by GO (data not shown). Next, fluorescence (Ex. 380 nm, Em. 445 nm) was measured with an Infinite 200 PRO plate reader (Tecan, Vienna, Austria). The software i-control 1.10 was employed.

### 2.5. Human Material

All experiments using human material were performed in accordance with the institutional guidelines and the agreement of the Ethical Committee of the University Hospital Erlangen (permit # 193 13B).

### 2.6. Detection of Complement Activation

Heparinized (20 U/mL) venous blood was taken from normal healthy donors (NHD) and centrifuged for 10 min at 2000× *g*. Next, plasma was collected and centrifuged for 10 min at 2000× *g*. The obtained platelet poor plasma (PPP) was incubated with graphene derivatives (200 µg/mL) for 1 h at 37 °C. Cobra venom factor (CVF; 8 i.U./mL) (Quidel, San Diego, CA, USA) or PPP alone were used as positive and negative controls, respectively. Next, graphene was sedimented by centrifugation for 15 min at 300× *g*. Samples were diluted 1:20 in denaturating sample buffer. SDS-PAGE (4%–20% bis-acrylamide) and western blot analysis were performed. Briefly, after protein transfer (2 h, 80 V), the Immobilon-PSQ PVDF membrane was blocked for 1 h in 5% milk at room temperature. Next, the membrane was incubated with primary rabbit anti-human C3 antibody (Abcam, Cambridge, UK) and secondary goat anti-rabbit antibody conjugated with horseradish peroxidase (Biozol Diagnostica, Eching, Germany). Detection was performed with a Celvin S scanner (Biostep, Burkhardtsdorf, Germany) and the software SnapAndGo 1.

### 2.7. Measurement of Plasma Coagulation

PPP was prepared from venous blood anticoagulated with sodium citrate as described above. Next, PPP was incubated with graphene derivatives (50 µg/mL) for 30 min at 37 °C and placed in pre-warmed cuvettes with a metal ball inside. Activated partial thromboplastin time (aPTT) or thrombin time (TT) were measured on an MC4plus macro coagulometer (Merlin medical, Lemgo, Germany) after the addition of aPTT reagent and calcium chloride (CaCl_2_) or thrombin reagent (all from DiaSys Diagnostic Systems GmbH, Holzheim, Germany), respectively.

### 2.8. Detection of Platelet Aggregation

Heparinized (20 U/mL) venous blood of NHD was centrifuged for 30 min at 200× *g* at room temperature. The platelet rich plasma (PRP) was incubated with graphene derivatives (50 µg/mL) for 15 min at 37 °C with continuous shaking. Photos were taken on Microscope Axiovert 25 (Zeiss, Oberkochen, Germany) by Nikon D700 reflex camera and were processed using Adobe Photoshop CS5.

### 2.9. Measurement of Plasma Membrane Stability

Heparinized (20 U/mL) venous blood of NHD was used and hemoglobin content was adjusted (2 mg/mL). Diluted blood was incubated with graphene derivatives (50 µg/mL) for 3 h at 37 °C and mixed every 30 min. Triton X-100 (1%) and phosphate buffer saline (PBS) were used as positive and negative controls, respectively. Next, erythrocytes were sedimented by centrifugation for 15 min at 800× g and collected supernatants were incubated with Drabkin’s reagent (Sigma-Aldrich, Germany) in 1:1 ratio in 96-well plate for 5 min at 60 °C. Detection at 590 nm was performed on an Infinite^®^ 200 PRO plate reader and the software i-control 1.10.

### 2.10. Detection of lactate dehydrogenase (LDH) release

The Pierce™ LDH Cytotoxicity Assay Kit (Thermo Fisher Scientific, Waltham, MA, USA) was used according to the manufacturer’s instructions. Human peripheral blood mononuclear cells (PBMC) were isolated from heparinized (20 U/mL) venous blood of NHD by density gradient centrifugation (Lymphoflot, Bio-Rad, city, state if USA, country). Cells (2 × 10^5^ cells/well) were incubated with graphene derivatives (50 µg/mL) in 12-well flat-bottom plates for 4 h at 37 °C and 5% CO_2_. The supernatants of PBMC and treated B16F10 melanoma cells were collected at indicated time points and centrifuged for 5 min at 300× *g*. Absorbance was measured at 492 nm and 650 nm reference wavelengths on an Infinite^®^ 200 PRO plate reader, and the software i-control 1.10 was used.

### 2.11. Detection of Neutrophilic Degranulation

Human polymorphonuclear neutrophils from NHD were isolated from the buffy coat after by density gradient centrifugation on Lymphoflot and subsequent erythrocyte lysis. Cells (4 × 106 cells/well) were incubated with graphene derivatives (50 µg/mL) in 96-well flat-bottom plates for 4 h at 37 °C and 5% COEthanol (20% final concentration) or cells alone were used to determine maximal or baseline degranulation levels, respectively. The enzymatic activity of myeloperoxidase (MPO) was detected with 3,3’,5,5’-tetramethyl benzidine (TMB) substrate set (Thermo Fisher Scientific, USA) according to the manufacturer’s instructions. The addition of H_2_SO_4_ (25%) stopped the reaction. Next, supernatants were collected after centrifugation for 5 min at 300× g and absorbance was measured at 450-nm and 620-nm reference wavelengths on an ELISA Microplate Reader (Tecan, Vienna, Germany) and the software Magellan 7.1 SP1.

### 2.12. Detection of Neutrophil Extracellular Traps (Nets)

Neutrophils (1 × 10^6^ cells/well) were incubated with graphene derivatives (50 µg/mL) in 96-well flat-bottom plates for 4 h at 37 °C and 5% CO_2_ with or without serum. Phorbol-12-myristat-3-acetat (PMA) (10 ng/mL) (Sigma-Aldrich, Germany) or cells alone were used as positive and negative controls, respectively. Next, deoxyribonuclease I (DNase I) (100 µg/mL) (Sigma-Aldrich, Germany) was added to each well for 30 min at 37 °C and 5% COThe activity of released neutrophil elastase (NE) was measured in supernatants after the addition of the fluorogenic substrate MeOSuc-AAPV-AMC (Santa Cruz Biotechnologies, Dallas, TX, USA) at 100 µM initial concentration. Fluorescence was measured on Infinite^®^ 200 PRO plate reader at 37 °C (Ex.360 nm, Em.465 nm) and the software i-control 1.10.

### 2.13. Quantification of Cell Migration

To verify chemotaxis, 96-well Chemo-Tx plate (NeuroProbe, Gaithersburg, MD, USA) was used according to the manufacturer’s instructions. Briefly, THP-1 cells (1 Mio/mL) were incubated with graphene derivatives at the indicated concentrations for 2 h at 37 °C and 5% CO_2_ on the upper part of Chemo-Tx plate (inserts with 5-µm pores). The migration towards 50 ng/mL of monocyte chemoatracctant protein-1 (MCP-1) (ImmunoTools, Friesoythe, Germany) present in the lower compartment of the plate during 1 h was quantified by counting cells collected from the lower compartment with a Gallios flow cytometer (Beckman Coulter, Krefeld, Germany) and the software Kaluza 2.Wells without MCP-1 served as a negative control. 

### 2.14. Phagocytosis

Macrophages were differentiated by stimulation with PMA (5 ng/mL) of THP-1 cells for 48 h at 37 °C and 5% CO_2_ on LabTek2 Chamber Slides (Thermo Fisher Scientific, USA) coated with Poly-L-Lysine hydrobromide (Sigma-Aldrich, Germany) (0.1 mg/mL). Macrophages were incubated with graphene derivatives (50 µg/mL) for 2 h at 37 °C and 5% COLive cell imaging was conducted on the BZ-X700 microscope (Keyence Corporation, Osaka, Japan) at 37 °C and 5% COImages were processed in Adobe Photoshop CS5 and time lapsed animations were composed with the Microsoft GIF animator.

### 2.15. Gamma Irradiation (X-rays)

B16F10 cells were exposed to 10 Gy of ionizing irradiation (120 kV, 22.7 mA; GE Inspection Technologies, city, Germany).

### 2.16. Graphene-Induced Hyperthermia (GIHT)

B16F10 melanoma cells derived from the C57BL/6 mouse were purchased from the ATCC (#CRL-6475). The cells (2 × 10^5^ cells/well) were seeded in 24-well flat-bottom culture plates, and graphene derivatives (50 µg/mL) were placed in transwell inserts with 0.4-µm pores. Next, wells were exposed to near-infrared irradiation (960 nm) for 1 h at a power density of 2 W/cm^2^ applied by Hydrosun^®^750 (Hydrosun Medizintechnik, Müllheim, Germany). The temperature of the medium in the lower compartment was recorded every 10 s with a Voltcraft K204 Thermometer (Voltcraft, Wollerau, Switzerland).

### 2.17. Statistics

Statistical analysis was performed by GraphPad Prism (version 7.0) software. As statistically significant, the *p*-values ≤ 0.05 were considered.

## 3. Results

### 3.1. Physicochemical Characterization of Graphene Oxide Nanosheets and Its Derivatives

We employed Fourier-transform infrared (FTIR) spectroscopy and Ultraviolet-Visible (UV/VIS) spectroscopy to analyse graphene derivatives. We observed that the absorption peak of GO was at 230 nm, whereas in rGO and rGO-PEG, it is shifted to 273 nm due to the partial restoration of the electronic conjugation (Figure 1A). The O–H stretching vibrations bands in the ~3430 cm^−1^ region of all FTIR spectra are indicative of absorbed water (Figure 1B). The bands at 1728 cm^−1^, 1610 cm^−1^, and 800–1500 cm^−1^ correspond to the vibration of C=O, C=C, and single (C–O) functions, respectively. The success of the PEGylation step is confirmed by the presence of the characteristic stretching bands at 2850 cm^−1^, at 1640 cm^−1^ and in the range of 800–1500 cm^−1^ corresponding to C–H, CO–NH, and C–O functions, accordingly. The dominant band at ~1570 cm^−1^ in rGO-PEG suggests that the aromatic structures (C=C stretching modes) were not affected during the process of functionalization with PEG. 

TEM images of GO, rGO, and rGO-PEG nanosheets show a sheet-like structure for GO and rGO. Contrarily, rGO-PEG displays a spherical morphology of 100 nm in diameter. All three graphene derivatives have a negative surface charge, indicating the presence of notably carboxylic functions and ensuring water solubility (Figure 1C,D). The solubility of rGO is lower compared to GO and rGO-PEG and has the tendency to form aggregates in biological media. Contrarily, the presence of the hydrophilic poly (ethylene glycol) chains in rGO-PEG results in improved solubility and stability up to 200 µg/mL (Figure 1E).

### 3.2. Sterility of Graphene Oxide Nanosheets and Its Derivatives 

When synthesized, nanomaterials are easily contaminated with endotoxin from the Gram-negative bacterial cell membrane. Endotoxin is a potent inducer of the immune system and may mask the real biological effects of nanomaterials. Therefore, we first screened graphene derivatives for microbial and endotoxin contamination. We performed a quick agar sterility test where GO, rGO, and rGO-PEG nanosheets were streaked on LB agar plates and incubated for 72 h at 37 °C, conditions allowing bacterial growth. As a positive control, we used *Escherichia coli* (strain DH5α). All of the GO nanosheets and its derivatives were free of bacterial contamination (Figure 1F). Next, we performed the fluorescent Limulus amoebocyte lysate assay that allows quantitative measurement of endotoxin levels. The recombinant Factor C is a synthetic endotoxin receptor that, upon binding to ligands, gets activated and converts the synthetic fluorogenic substrate into a fluorescent product. All tested nanomaterials revealed levels of endotoxin smaller than 0.053 EU/mL (Figure 1G). The endotoxin spiking control indicated no optical interference of the reduced forms of GO (rGO and rGO-PEG) with the colorimetric method (data not shown). These results show that all rGO derivatives can be safely used under regular culture conditions including immune cells.

### 3.3. Hemocompatibility of Graphene Oxide Nanosheets and Its Derivatives

Red blood cells are the most abundant cells in circulation. Their principal function is to transport oxygen and carbon dioxide to and from tissues, respectively. Uncontrolled erythrocyte destruction leads to the toxic discharge of hemoglobin into the plasma. The release of hemoglobin from reducing intracellular milieu can lead to multiple pathophysiological events such as an immediate increase of blood pressure [14], pro-oxidative toxicity in veins, and mechanical obstruction of renal tubules [15,16]. Since small-sized nanoparticles can disrupt lipid bilayers [17,18], we performed quantitative colorimetric determination of free hemoglobin of red blood cells co-incubated with graphene nanosheets. The levels of extracellular hemoglobin in the supernatants of erythrocytes exposed to GO nanosheets and its derivatives were not significantly different from the negative control values. Therefore, none of the graphene derivatives leads to red blood cells lysis (Figure 2A).

Complement proteins are one of the oldest components of the immune system. The complement system is involved in the opsonization and lysis of membrane coated microorganisms, chemotaxis of leukocytes, and removal of immune complexes, and some of its cleaved proteins function as anaphylatoxins (e.g., C3a and C5a). Uncontrolled stimulation of this system results in the release of anaphylatoxins which consequently can lead to death [19]. It is known that nanomaterials are potent inducers of the complement system and that the physical, chemical, and optical properties are key players in this process [20,21]. We performed western blot analysis to determine if graphene derivatives are capable to activate the complement system. Three western blot analyses of human plasma were performed. As a positive control (PC) of complement activation, we used Cobra Venom Factor (CVF), and PBS was used in the negative control (NC). The complement system split product detected in PC was absent in samples where GO, rGO, and rGO-PEG nanosheets were added. We, therefore, assume that graphene nanosheets do not activate the complement system at the concentration of 50 µg/mL (Figure 2B).

Proteins of the coagulation system, once activated, act in a cascade of enzymatic events that lead to the formation of a stable clot necessary to avoid blood loss after injury. However, it can also contribute to the partial or total occlusion of vessels by thrombi. We investigated whether GO, rGO, and rGO-PEG nanosheets have any influence on plasma coagulation in vitro. After separating plasma from blood of three NHD, plasma samples were co-incubated with GO, rGO, or rGO-PEG. Clot formation was detected mechanically by a coagulometer. The activated partial thromboplastin time (aPTT) and thrombin times of plasma containing GO and its derivatives are not significantly different from those of the fresh NHD plasma. Thus, tested GO nanosheets and its derivatives do not trigger the coagulation cascade of human plasma (Figure 2C,D).

Platelets are a subset of blood cells that may adhere to nanomaterials and may initiate the formation of potentially deadly clots [22]. We incubated platelet rich plasma with 50 µg/mL of GO nanosheets. Adenosine diphosphate (ADP) was used at the concentration of 2 µmol/mL as the aggregation stimulator. The activation of platelets by ADP resulted in a formation of microscopically visible aggregates (blue arrows) independently of the presence of GO nanosheets and its derivatives (Figure 3). Furthermore, GO nanosheets and its derivatives did not induce platelet aggregation.

### 3.4. Interaction of Graphene Oxide Nanosheets and Its Derivatives with Leukocytes

We evaluated the viability of leukocytes by exposing freshly isolated PBMCs to GO nanosheets and its derivatives in complete culture medium for 24 h. The lactate dehydrogenase (LDH) levels in supernatants were used as a measure of cytotoxicity. The coupled redox reaction in which tetrazolium salt is reduced to a red formazan allows the colorimetric measurement of LDH at 490 nm. No significant LDH release was detected during the co-incubation of GO, rGO, and rGO-PEG with PBMCs (Figure 4A). These results indicate that graphene oxide nanosheets and their derivatives have no influence on the viability of leukocytes at the concentration of 50 µg/mL. 

Once nanomaterials populate the tumor tissue, they will encounter cells of the monocytic/macrophage lineage that are continuously patrolling tissues. In order to assess the interaction of GO nanosheets and its derivatives with THP-1 monocytes, we set up two-chamber migration assays. Monocytes were seeded on trans well inserts (5-µm pore size) with GO nanosheets and its derivatives and were allowed to migrate towards chemoattracting MCP-1 (monocyte chemoattractant protein-1) in the lower compartment. Migrated cells were collected from the lower chamber, fixed, and counted by flow cytometry. The migration of monocytes was significantly affected by elevated concentrations of GO (200 and 400 µg/mL; Figure 4B), rGO (400 µg/mL; Figure 4C), and rGO-PEG (200 and 400 µg/mL; Figure 4D), indicating an interaction of the phagocytic cells with the nanosheets. Monocytes probably engulfed the nanosheets, secreted MCP-1, and disrupted the chemoattractant gradient that drove the migration.

In order to closer evaluate the engulfment of GO nanosheets and its derivatives by phagocytic cells, we exposed the differentiated macrophages to 50 µg/mL of GO, rGO, and rGO-PEG and monitored the culture with the bright field microscope. All three types of GO nanosheets accumulate on the bottom of the wells and are actively cleared by crawling macrophages (Appendix A, macrophages were highlighted with a red line).

Neutrophils are the most numerous leukocytes and the first cells that appear at the site of inflammation to initiate the response against invaders. Since it was reported that particulate matter can lead to the activation of neutrophils [17,23,24], we investigated whether GO nanoderivatives can induce degranulation and/or formation of neutrophil extracellular DNA traps (NETs) in neutrophils. By measuring the activity of one of the major granular proteins (MPO) released by neutrophils, we measured degranulation. The co-culture of neutrophils with GO nanosheets and its derivatives did not trigger degranulation (Figure 4E). NET formation was detected by the release of enzymatically active neutrophil elastase after digestion with DNase as described in Reference [25]. With this method, we demonstrated the presence of NETs after stimulation of neutrophils in serum-free medium but not in serum-containing medium (Figure 4F,G). From these results, we assume that, in in vivo conditions, GO nanosheets and its derivatives will not cause the activation of neutrophils.

### 3.5. Graphene-Induced Hyperthermia (GITH) and Tumor Cell Death

The excellent light to heat conversion of graphene derivatives due to the increased light absorbance in the near-infrared (NIR) gained attention with the development of thermotherapies, especially in cancer treatment. GO nanosheets and its derivatives in complete culture medium were exposed simultaneously and intermittently to NIR (960 nm) at the power density of 225 mW/cm^2^ until one of the wells reached 43 °C. We observed a rapid temperature increase in the cases of GO, rGO, and rGO-PEG when compared to the medium control (Figure 5A). rGO and rGO-PEG nanosheets showed superior photothermal heating than that of GO. This demonstrates that especially rGO and rGO-PEG nanosheets are excellent light to heat converters of NIR light with good heat transfer to the culture medium. In this fashion, precise temperature conditions can be achieved in the close vicinity of the GO nanosheets and its derivatives without affecting neighboring zones.

In order to confirm cell death in B16F10 melanoma cells, we detected the activity of LDH in the supernatants 24 h after the GIHT, X-ray treatment, and the combination of both. The hyperthermia (43 °C) induced by rGO and rGO-PEG alone or in the combination with X-ray leads to increased LDH activity in comparison to X-radiation applied without prior thermal treatment (Figure 5B–F). The temperature reached with GO (below 41 °C) resulted in LDH levels similar to those detected in medium without nanosheets. This indicates that 41 °C hyperthermia is not sufficient to induce cell membrane damage and LDH release from tumor cells. Additionally, we observed that the release of LDH was significantly decreased in X-ray treatment when the cells were co-incubated with GO in comparison to cells in medium alone.

## 4. Discussion

We have investigated herein how the reduction and PEGylation of GO nanosheets account for their biocompatibility, light to heat conversion of NIR light, and their potential application in the induction of localized hyperthermia as coadjuvant in anticancer therapies. First, we aimed to confirm the sterility of GO nanosheets and its derivatives to exclude the presence of bacteria or endotoxin traces, potent stimulators of the immune system. These two elements can mask the real biological effect of nanomaterials [26,27,28,29]. In a quick agar sterility test, we confirmed that GO, rGO, and rGO-PEG nanosheets are free of bacterial contamination and that the levels of endotoxin measured by the Limulus amoebocyte lysate assay resulted below of the maximum amount accepted by the FDA for the employment of nanomaterials in medicine [30]. These results confirm the sterility of GO nanocomposites and enable us to further conduct biological tests.

Nanomaterials can be administered into living organisms through multiple routes, oral [31,32], pulmonary [33,34], transdermal [35,36], intraperitoneal [32], or intravenous [37,38]. The administration route influences the toxicity of the nanocomposites. For example, inhalation and direct distribution of GO in the lung results in increased ROS production that drives severe inflammation and lung injury [39]. In contrast, intraperitoneal administration of GO and GO-PEG results in endocytosis, and despite their long retention, no obvious signs of toxicity are detected. The oral route leads to the fast removal of GO-PEG that cannot be absorbed in the gastrointestinal tract [40]. The final goal of this study is to collect evidence that ensures the safe delivery of GO nanosheets to tumor tissues where its photothermal conversion properties can be exploited to induce fine-tuned hyperthermia and to radio-sensitize cancerous cells. For this purpose, we recommend the intravenous administration of GO nanosheets and its derivatives, allowing the passive accumulation of nanomaterials in the tumor by the EPR effect [41,42,43]. Yang et al. reported that PEGylated graphene nanosheets exhibit high accumulation in the tumor tissue [44]. The same research group demonstrated that rGO-PEG (65 nm) and ultrasmall rGO-PEG (27 nm) were characterized by prolonged half-time after intravenous injection in comparison to ultrasmall GO-PEG (23 nm) [45]. This feature allows successive passages of the nanosheets through the tumor increasing its distribution in the cancer tissue via the EPR effect.

In line with the recommended route of administration, we investigated the effect of GO nanosheets on various components of the bloodstream. All the experiments were conducted at a maximum concentration of 50 µg/mL since it was demonstrated that the toxicity of GO is dose-dependent and increases with the concentrations higher than 50 µg/mL [46]. We show that sole GO nanosheets and its derivatives do not elicit permeabilization of the plasma membrane, cell death, activation of the complement system, aggregation of platelets, or coagulation of the plasma. In addition, they do not interfere with the degranulation of neutrophils or with the migration of monocytes towards the chemoattractant. 

NET formation was detected in the response of neutrophils exposed to GO and rGO-PEG exclusively under serum-free conditions. The process was reportedly abolished by the addition of serum proteins similar to other stimuli [47]. Munoz et al. showed that initiation of NET release depends on the size of nanomaterial [17]. They observed that the co-incubation of neutrophils with small hydrophobic nanoparticles (10-nm diamonds and 40-nm polystyrene beads) leads to the detection of structures resembling PMA-induced NETs. NET formation was absent when larger nanoparticles were used (1000-nm diamonds and 1000-nm polystyrene beads). We, therefore, suggest that observed NET ejection was elicited by a smaller size of GO and rGO-PEG: 570 ± 19 nm and 289 ± 4 nm, respectively. Consistently larger rGO (2738 ± 301 nm) nanosheets were inert to neutrophils in serum-free conditions. Employing the intravenous path of administration protein-rich conditions is granted; therefore, we conclude that the formation of NETs elicited by graphene derivatives will be inhibited in this milieu. Importantly, upon contact with body fluids, nanocomposites are covered by the proteins of plasma [48,49] that adsorb with high (hard corona) or low (soft corona) affinity to the nanomaterial [50]. Each nanomaterial is coronated with different proteins depending on its physicochemical properties (size, shape, functionalization, composition, and charge), the type of physiological compartment (blood, interstitial fluid, and cytoplasm), and the time of exposure. This process defines the biological identity of nanocomposite and influences important parameters, such as cytotoxicity [51], biodistribution, or cellular uptake [52,53,54,55]. Therefore, the inhibition of NET formation can be explained either by coronation or by the effect of serum proteins in the stabilization of membranes in neutrophils [47]. Taken together, GO nanosheets can be considered biocompatible at the concentration of 50 µg/mL in biological fluids containing proteins.

Another important aspect of biocompatibility is biodegradability. The understanding of the pathways leading to the removal of nanomaterials from the organism is of major importance for their final application and is a required issue for the clearance of use in clinical trials. Nanocomposites, firstly considered as structurally persistent, were shown to be degraded by oxidative enzymes, among others, peroxidases [56,57,58,59]. It was reported that single layer and few layers graphene is degraded by human MPO, which is released by neutrophils [60]. In our in vitro settings, we did not detect degranulation of neutrophils in the presence of graphene nanoderivatives. Nevertheless, artificial enzymes, such as DNAzymes mimicking peroxidase activity, were shown to degrade GO [61] and could be alternatively employed. To note, granulocytes are not the only cells that express peroxidases. Macrophages, monocytes, and monocyte-derived cells were demonstrated to possess and secrete peroxidase [62]. We showed that, independently on the type of used nanosheets (GO, rGO, or rGO-PEG), THP-1-derived macrophages collect nanosheets that settle in their neighborhood. In a similar fashion, GO nanosheets and its derivatives will be cleared from the extracellular milieu of blood-perfused tissues. PEGylation, which was reported to decrease cellular uptake [63], had no visible effect on this process. Yang et al. utilized fluorescence labeling of PEGylated graphene nanosheets with cyanine 7 (Cy7) and followed their biodistribution in vivo [44]. The excitation/emission of Cy7 (ex. 754, em. 778 nm) fluorophore is in NIR field. In contrast, Lucifer yellow (ex. 428 nm, em. 540 nm) and pHrodo green (ex. 509 nm, em. 533 nm) are visible in the green light spectrum. Another available variant of pH-sensitive tracker is pHrodo red (ex. 560 nm, em. 585 nm) detected in the orange light spectrum. Hence, employing appropriate fluorophores, the tracking of the engulfment of graphene nanomaterials will allow delineating the precise biodistribution and biodegradation of graphene derivatives. 

Radiotherapy is an essential treatment option for the majority of patients bearing tumors [64]. However, radioresistance of some cancer cells results in the failure of this therapy [65]. In several studies, the nonresponsiveness to radiation was resolved with hyperthermia applied concomitantly with irradiation [64,66]. Therefore, we aimed to investigate the effect of GIHT and irradiation on the cell death of poorly immunogenic B16F10 melanoma cells. We chose a radiation dose of 10 Gy resembling the cumulative weekly dose of X-ray that patients are exposed to [67].

In the process of the search for the ideal graphene derivative for the hyperthermia application, we tested the heat conductivity of GO, rGO, and rGO-PEG nanosheets. We observed that rGO and rGO-PEG exhibited higher photothermal conversion efficacy in comparison to GO. This is in line with another report showing superior thermal conduction of rGO over GO [68]. Contrarily to other reports [63], the functionalization of rGO with PEG did not change this feature. Importantly, the temperature reached by the medium in the absence of graphene nanoderivatives is in a range of mild fever (≤ 39 °C). Therefore, we conclude that the energy applied will not cause damage to the surrounding normal noncancerous tissues that are not flooded with GO nanosheets and its derivatives and will allow localizing the heating in the tumor area only. Additionally, the temperature amplitude observed for rGO and rGO-PEG was higher that of GO. This demonstrates the great capacity of heat conductivity of rGO and rGO-PEG that absorb the energy from the NIR irradiation and give it away in the form of heat with high efficiency. This feature can be determinant in shortening the time that patients need to heat up and cool down the area of treatment. The possibility to reach various temperatures using GO or rGO(-PEG) allows investigating the difference between two levels of GIHT. The hyperthermia induced by rGO and rGO-PEG (heat delivered in pulses of 42–43 °C for 1 h) alone or in the combination with X-ray (10 Gy) led to significantly increased cell death when compared to radiation alone. 

The physicochemical analysis of all three types of GO nanosheets revealed that, due to its spherical shape and increased stability in aqueous suspension, rGO-PEG is highly biocompatible, and it can be successfully utilized to induce localized hyperthermia of solid tumors by NIR. The great heat conductivity of rGO-PEG allows fine-tuning of temperature at 42–43 °C by applying short pulses of NIR. 

Gamma irradiation induces irreversible double-strand DNA breaks that trigger the execution of the apoptotic routine. In the in vivo situation, dying tumor cells are sensed by the immune system and transmit predominantly tolerogenic signals [69,70,71]. The rationale for hyperthermia accompanying radiotherapy is that dying cells progress to more immunogenic forms of death like secondary necrosis [72,73]. We conclude that rGO-PEG is biocompatible and nontoxic and can be used for intravenous application to induce localized hyperthermia in combination with gamma irradiation to enhance tumor cell death. It remains to determine whether the type of cell death triggered by the combination of both treatments would result in more effective antitumor immune responses.

## Figures and Tables

**Figure 1 cells-09-00776-f001:**
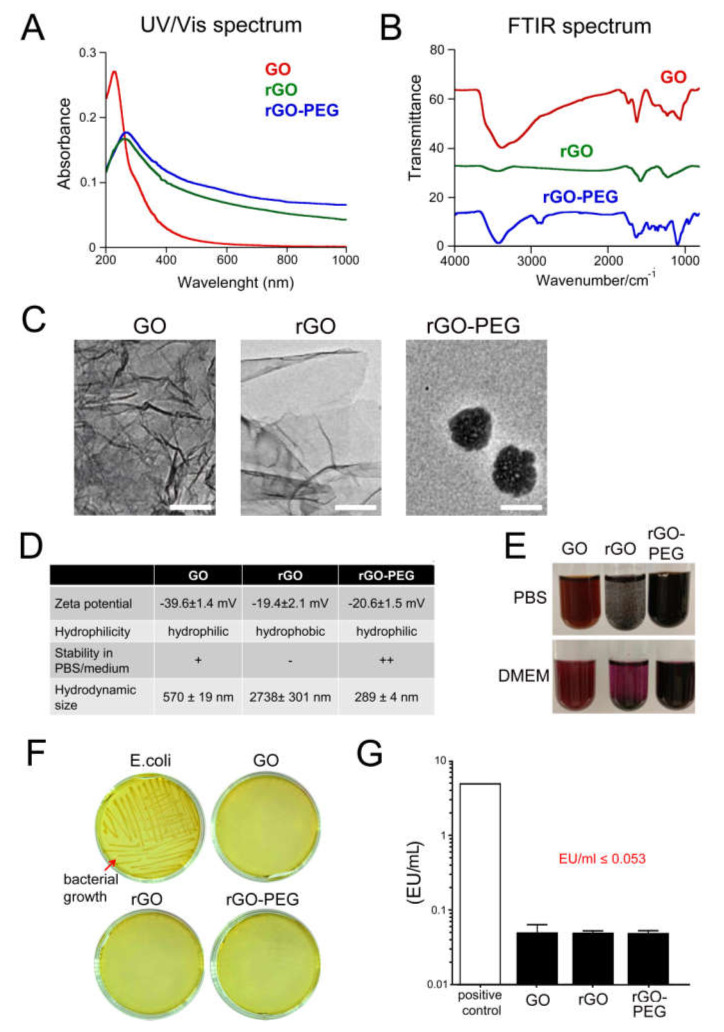
Characterization of graphene oxide (GO), reduced graphene oxide (rGO), and PEGylated rGO (rGO-PEG): (**A**) UV/Vis absorption spectra of graphene derivatives. The optical absorption was measured in 1 cm quartz cuvette on UV/Vis spectrophotometer, and the wavelength range of 200–1000 nm was used. (**B**) FTIR absorption spectra of graphene derivatives. The optical absorbance was measured on a Nicolet 8700 FTIR instrument (resolution 4 cm^−1^). The signal recorded from a pure KBr pellet was used as background noise. (**C**) Representative transmission electron microscope (TEM) images of graphene derivatives. rGO-PEG was adapted from [12,13]. Scale bar, 100 nm. (**D**) Physicochemical characterization of graphene derivatives. (**E**) Representative photographs of graphene derivatives diluted in PBS or DMEM as depicted. (**F**) Determination of bacterial growth on Luria broth (LB) agar plates covered with 50 µg/mL of graphene derivatives. Positive control: *Escherichia coli* (strain DH5α), red arrow. Scale bar, 2 cm. (**G**) Quantitative measurement of bacterial endotoxin levels in graphene derivatives. UV/Vis, Ultraviolet-Visible; FTIR, Fourier-transform infrared spectroscopy; GO, graphene oxide; rGO, reduced graphene oxide; rGO-PEG, PEGylated reduced graphene oxide; LB, Luria broth; *E. coli*, *Escherichia coli*; EU, endotoxin units.

**Figure 2 cells-09-00776-f002:**
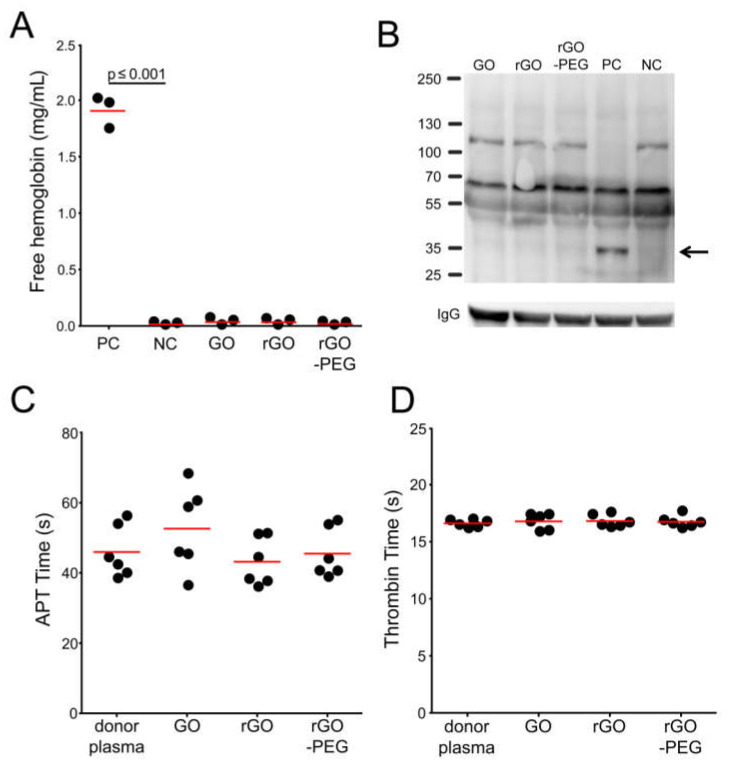
Hemocompatibility of GO, rGO, and rGO-PEG: (**A**) Detection of hemolysis in the presence of graphene derivatives at 50 µg/mL with normal healthy donor (NHD) venous blood. Positive (blood in PBS + 1% Triton X-100) and negative (blood in PBS) controls were employed. Each dot represents a mean value of one donor. One-way analysis of variances with Tukey’s multiple comparison test was used. (**B**) Determination of the activation of the complement system in fresh human plasma co-incubated with 50 µg/mL of graphene derivatives. Positive (plasma + Cobra venom factor (CVF); PC) and negative (plasma + PBS; NC) controls were employed. The complement activation product is depicted by a black arrow. C3 (α chain) is 115 kDa; C3 split products are around 43 kDa. The molecular weight marker is shown on the left. The image is representative of three independent experiments. Measurement of plasma coagulation time: Fresh human plasma was co-incubated with 50 µg/mL of graphene derivatives for 30 min at 37 °C. Plasma coagulation was induced by the addition of calcium chloride and activated partial thromboplastin time (aPTT) reagent or thrombin, and aPTT (**C**) or thrombin time (**D**) were recorded, respectively. One-way analysis of variances with Dunnett’s multiple comparison test was applied. Technical duplicates were used, *n* = GO, graphene oxide; rGO reduced graphene oxide; rGO-PEG, PEGylated reduced graphene oxide; CVF, cobra venom factor; aPTT, activated partial thromboplastin time.

**Figure 3 cells-09-00776-f003:**
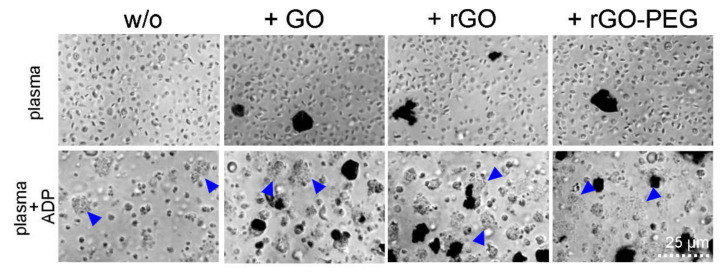
The effect of GO, rGO, and rGO-PEG on platelet activation: Graphene nanoderivatives were co-incubated with platelet rich plasma at 50 µg/mL. Platelet activation was induced by the addition of 2 µmol/mL of adenosine diphosphate (ADP). Platelet aggregates are pointed by blue arrows. One representative picture reflects the results of three independent experiments. GO, graphene oxide; rGO reduced graphene oxide; rGO-PEG, PEGylated reduced graphene oxide; ADP, adenosine diphosphate.

**Figure 4 cells-09-00776-f004:**
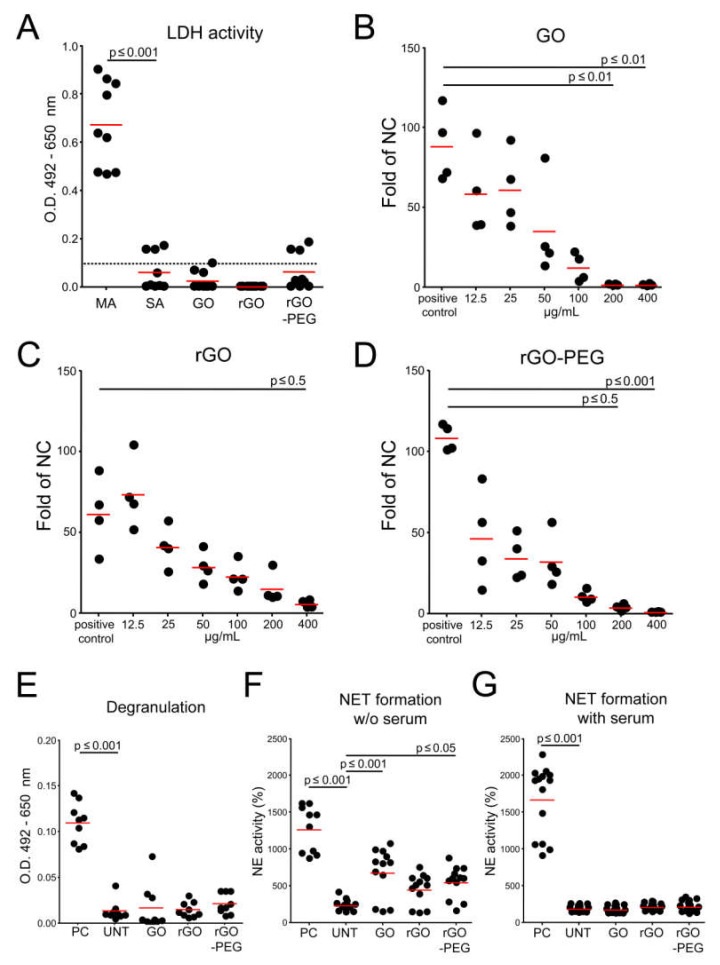
Reaction of leukocytes towards GO, rGO, and rGO-PEG in vitro: (**A**) Viability of peripheral blood mononuclear cells (PBMCs). Graphene derivatives (50 µg/mL) were co-incubated with freshly isolated PBMCs, and the lactate dehydrogenase (LDH) activity was measured in supernatants. Maximum LDH activity (PBMCs + lysis buffer; MA) and spontaneous LDH activity (PBMC + water; SA) were assessed. LDH activity detected in supernatants of untreated PBMCs is shown as a black dotted line. One-way analysis of variances with Dunnett’s multiple comparison test were applied. Technical triplicates are depicted for each donor, *n* = 3. (**B**–**D**) The THP-1 monocytic cell line was used for chemotaxis assays. Monocytes were co-incubated with various concentrations of graphene derivatives in trans-well inserts (upper compartment) and the chemokine monocyte chemoatracctant protein-1 (MCP-1) in the lower compartment. Results normalized to the negative control of technical quadruplicates are presented. One-way analysis of variances with Dunnett’s multiple comparison test was applied. (**E**) Determination of neutrophilic degranulation was measured by the myeloperoxidase (MPO) activity released to the supernatants. Freshly isolated neutrophils were co-incubated with 50 µg/mL of graphene derivatives in a complete medium. The positive control (neutrophils + ethanol; PC) was used. (**F**,**G**) Detection of Neutrophil Extracellular Trap (NET) formation in complete medium and in serum free medium, respectively: Freshly isolated neutrophils were co-incubated with 50 µg/mL of graphene derivatives. Positive control (neutrophils in medium + Phorbol-12-myristat-3-acetat (PMA); PC) and untreated cells (neutrophils in medium; UNT) were used. Neutrophil elastase (NE) activity was determined after DNase I digestion. (**E**–**G**) One-way analysis of variances with Dunnett’s multiple comparison test were applied. Technical duplicates or triplicates are presented, *n* = GO, graphene oxide; rGO, reduced graphene oxide; rGO-PEG, PEGylated reduced graphene oxide; LDH, lactate dehydrogenase; O.D., optical density; MCP-1, monocyte chemoattractant protein-1; MPO, myeloperoxidase; NET, neutrophil extracellular traps; NE, neutrophil elastase.

**Figure 5 cells-09-00776-f005:**
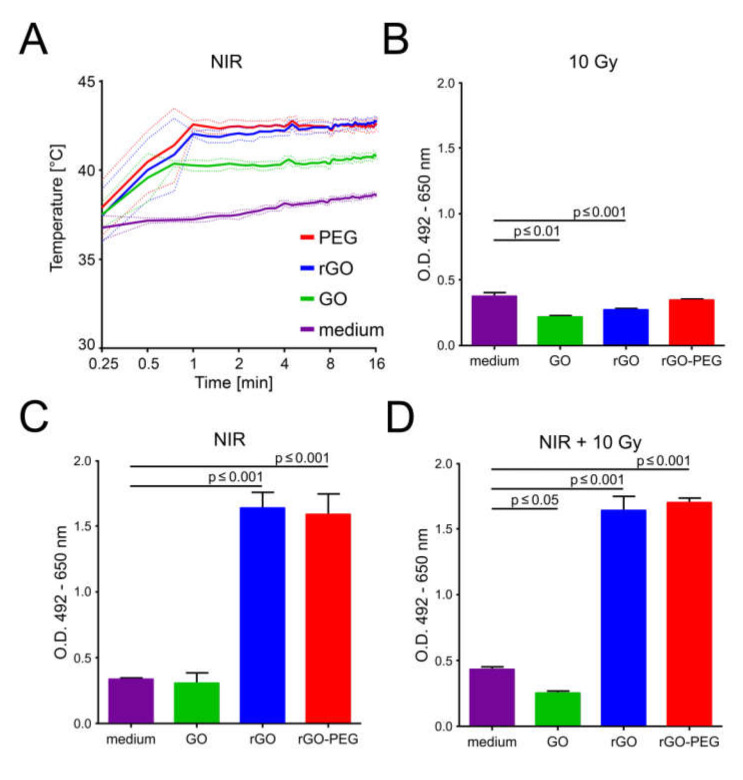
Viability of B16F10 melanoma cells after graphene induced hyperthermia (GIHT) alone or in combination with radiation: (**A**) Temperature curves of wells containing culture medium alone or with GO, rGO, or rGO-PEG during near-infrared (NIR) (960 nm) exposure. Dotted lines indicate standard deviations. (**B**–**D**) LDH release from B16F10 melanoma cells 24 h after indicated treatment. The basal LDH release of untreated cells was 0.042 ± 0.004 O.D. (not shown). Means standard error of the mean (SEM) are shown. One-way analysis of variances with Dunnett’s multiple comparison test were applied. GO, graphene oxide; rGO, reduced graphene oxide; rGO-PEG, PEGylated reduced graphene oxide; NIR, near-infrared; LDH, lactate dehydrogenase; Gy, Gray; O.D., optical density.

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
