# Peer review of "Graphene Oxide Nanosheets for Localized Hyperthermia—Physicochemical Characterization, Biocompatibility, and Induction of Tumor Cell Death"

_cells, 2020, doi:10.3390/cells9030776_

Round 1

Reviewer 1 Report

The manuscript describes on the details of the interaction of graphene oxide (GO) nanosheets, its derivatives reduced GO) and PEGylated rGO, with components of peripheral blood and evaluated its thermal conductivity to induce cell death by localized hyperthermia.

In addition, the authors also confirmed that the Biocompatible GO and its derivatives can be used Graphene-Induced Hyperthermia to elicit tumor cell death.

 The article is interesting, and the work is appropriate in scope and length for publication in the cells journal. However, few questions need to be addressed before its acceptance.

Characterizations of all the GO derivatives are not clear. Need to provide more details in supporting information (X-ray photoelectron spectroscopy and measurement of Fluorescence data, and microscopic images such as SEM,  EDS etc

Why hydrodymanic size of EGylated rGO are small than rGO and rGO is bigger than GO?

Please provide step by step synthetic schemes of GO, rGO, PEGylated rGO etc.

Author Response

Characterizations of all the GO derivatives are not clear. Need to provide more details in supporting information (X-ray photoelectron spectroscopy and measurement of Fluorescence data, and microscopic images such as SEM,  EDS etc.

R: We edited the description of the characterization of GO derivatives and added more details supporting the methods employed.

Why hydrodynamic size of PEGylated rGO are small than rGO and rGO is bigger than GO?

R: PEGylated rGO disperses nicely and results in decreased overall hydrodynamic size. In the case of rGO, aggregation is occurring and this increases the hydrodynamic size.

Please provide step by step synthetic schemes of GO, rGO, PEGylated rGO etc.

R: We added more details about the way the materials were synthesized as well as technical details about XPS, zeta potential, FTIR.

Reviewer 2 Report

A manuscript review

Graphene and its derivatives have emerged as promising materials with a potential to find applications in biomedicine. Therefore, it is crucial to further investigate interaction between human cells and these nanomaterials. In this manuscript results from comprehensive study of GO, rGO and rGO-PEG effect on blood cells have been presented. Cytotoxicity, plasma coagulation, platelet aggregation, cell migration, membrane stability, phagocytosis and immunity responses in cells treated with three graphene derivatives have been studied. Moreover, efficiency of hyperthermia therapy, radiotherapy and their combination using graphene-based nanomaterials on B16F10 melanoma cells have been evaluated. Results shows that GO, rGO and rGO-PEG cause no immunity response. What is more examined nanomaterials are biodegradable and can be phagocytosed by macrophages. In addition rGO and rGO-PEG transpired to be efficient agents in hyperthermia therapy. Research data presented in this article sheds more light on interactions between graphene-based nanomaterials and blood cells. However, it would seem to be appropriate to address some concerns, namely:

  1. Reduced graphene oxide purity: Using hydrazine hydrate as a reducing agent is widely used strategy to synthesize reduced graphene oxide (rGO). However this compound can strongly affect cell behavior. Therefore, it is very important matter to determine whether there are some hydrazine residues in rGO samples. Since characteristic bands for hydrazine hydrate in FTIR spectrum of examined samples can be easily missed, maybe it would be a good idea to use complementary analytical technique to ensure rGO purity.
  2. Particle size determination of rGO and GO: Size of nanoparticle is crucial factor in studying its interactions with cells. Dynamic light scattering (DLS) is well-established and widely-used technique for particle size determination. However it is difficult to obtain precise data with DLS for non-spherical particles like rGO and GO. It would be advised to use, along with DLS, another size determination technique.
  3. Abbreviations: Although most of abbreviations of chemical compounds are explained, some are not, like PMA. It could be confusing since more than one compound have this abbreviation.
  4. Figure 5: A plot with viability of cells when no therapy was applied could help to understand the impact of hyperthermia therapy and X-ray treatment on cells.
  5. Absorbance measurements: Based on Figure 1A it can be said that rGO and rGO-PEG are capable of absorbing UV/VIS light at 490nm, even after subtraction the absorbance for reference wavelength. Could this property of studied materials affect LDH test measurements?
  6. Figure 1G: There are no error bars for rGO and rGO-PEG. Was the experiment for these samples carried out once?
  7. APT time and monocyte migration: It is well known fact that in biological studies it is difficult to obtain reproducible data. However, especially as far as GO samples is concerned, it seems that values of APT time differs significantly between measurements (Figure 2C). Basing on this data could it be said that GO certainly does not affect APT time? Similar concern appear regarding chemotaxis assays (Figure 4B-D). In a discussion section it is written that examined nanomaterials does not interfere with monocyte migration. Figure 4B-D show that nanomaterials for concentration of 50 µg/ml affect monocyte migration.

Author Response

  1. Reduced graphene oxide purity: Using hydrazine hydrate as a reducing agent is widely used strategy to synthesize reduced graphene oxide (rGO). However this compound can strongly affect cell behavior. Therefore, it is very important matter to determine whether there are some hydrazine residues in rGO samples. Since characteristic bands for hydrazine hydrate in FTIR spectrum of examined samples can be easily missed, maybe it would be a good idea to use complementary analytical technique to ensure rGO purity.

R: rGO when formed is poorly soluble in water and any hydrazine traces are completely washed off in the following steps. Furthermore, the results shown in figures 2A and 4A confirm that GO-derivatives or any potential contaminant contained do not affect the integrity of biological membranes (Fig 2A) or the viability of leukocytes (Fig 4A).

  1. Particle size determination of rGO and GO: Size of nanoparticle is crucial factor in studying its interactions with cells. Dynamic light scattering (DLS) is well-established and widely-used technique for particle size determination. However it is difficult to obtain precise data with DLS for non-spherical particles like rGO and GO. It would be advised to use, along with DLS, another size determination technique.

R: Indeed, that is the reason we have made TEM images in addition. In the Figure 1C of the manuscript, the real size of the nanostructures can be appreciated.

  1. Abbreviations: Although most of abbreviations of chemical compounds are explained, some are not, like PMA. It could be confusing since more than one compound have this abbreviation.

R: We apologize for this omission. We added the meaning of this and other abbreviation at its first appearance in the manuscript.

  1. Figure 5: A plot with viability of cells when no therapy was applied could help to understand the impact of hyperthermia therapy and X-ray treatment on cells.

R: the readings of LDH release of untreated cells are now added in the figure 5 legend.

  1. Absorbance measurements: Based on Figure 1A it can be said that rGO and rGO-PEG are capable of absorbing UV/VIS light at 490nm, even after subtraction the absorbance for reference wavelength. Could this property of studied materials affect LDH test measurements?

R: The reviewer is right; GO-derivatives absorb UV/VIS light. Therefore, all LDH measurements were done in the supernatants as described in the methods section 2.10.

  1. Figure 1G: There are no error bars for rGO and rGO-PEG. Was the experiment for these samples carried out once?

R: These readings were done in duplicates. We apologize for this omission. The standard deviations of these bars were lost in the process of image composition. Figure 1G has been corrected.

  1. APT time and monocyte migration: It is well known fact that in biological studies it is difficult to obtain reproducible data. However, especially as far as GO samples is concerned, it seems that values of APT time differs significantly between measurements (Figure 2C). Basing on this data could it be said that GO certainly does not affect APT time? Similar concern appear regarding chemotaxis assays (Figure 4B-D). In a discussion section it is written that examined nanomaterials does not interfere with monocyte migration. Figure 4B-D show that nanomaterials for concentration of 50 µg/ml affect monocyte migration.

R: The readings of activated partial thromboplastin time (APTT) are indeed quite variable even in the case of plasma alone. Therefore we employed and additional coagulation parameter to confirm the absence of effect. In the case of migration assays, there is a non-significant reduction of the migration also at lower concentrations of GO derivatives. This can also be explained by the engulfment of the particles by the cells as discussed in the manuscript. This event is not to be accounted as negative effect on the immune system but more as a reflection of the clearance function of the monocytic system encountering the GO derivatives. This is  further supported in the animations provided as supplementary material which were also made with 50µg/ml suspensions. This aspect was elaborated in the discussion.

Round 2

Reviewer 2 Report

Regarding authors' responses on my comments, I accept them and recommend to publish this revised manuscript.